# Effect of Copper Sulphate and Cadmium Chloride on Non-Human Primate Sperm Function In Vitro

**DOI:** 10.3390/ijerph18126200

**Published:** 2021-06-08

**Authors:** Farren Hardneck, Charon de Villiers, Liana Maree

**Affiliations:** 1Department of Medical Bioscience, University of the Western Cape, Private Bag X17, Bellville 7535, South Africa; farren@hardneck.co.za; 2PUDAC-Delft Animal Facility, South African Medical Research Council, Cape Town 7505, South Africa; charon.devilliers@mrc.ac.za

**Keywords:** sperm motility, sperm vitality, hyperactivation, acrosome integrity, vervet monkey, chacma baboon, rhesus monkey

## Abstract

In order to address the large percentage of unexplained male infertility in humans, more detailed investigations using sperm functional tests are needed to identify possible causes for compromised fertility. Since many environmental and lifestyle factors might be contributing to infertility, future studies aiming to elucidate the effect of such factors on male fertility will need the use of appropriate research models. The current study aimed to assess the effects of two heavy metals, namely copper sulphate, and cadmium chloride, on non-human primate (NHP) sperm function in order to establish the possibility of using these primate species as models for reproductive studies. Our combined results indicated that the functionality of NHP spermatozoa is inhibited by the two heavy metals investigated. After in vitro exposure, detrimental effects, and significant lowered values (*p* < 0.05) were obtained for sperm motility, viability and vitality, acrosome intactness, and hyperactivation. These metals, at the tested higher concentrations, therefore, have the ability to impair sperm quality thereby affecting sperm fertilizing capability in both humans and NHPs.

## 1. Introduction

The decrease in human male fertility (lower sperm count and sperm motility) during the last few decades may partly be due to exposure to environmental [1] and occupational contaminants along with certain lifestyle practices [2]. Several lifestyle factors are associated with male infertility, including cigarette smoking, alcohol intake, use of illicit drugs, obesity, psychological stress, and dietary practices [3]. Humans are exposed to various types of environmental contaminants at different stages of their life and there has been a growing concern regarding the deleterious effects on the developing male reproductive system [4]. Environmental contaminants, such as plastics, pesticides, pharmacological agents, and heavy metals are classified as gonadotoxins and are contributing factors in unexplained (idiopathic) male infertility [5]. Growing evidence suggests that persistent contaminants may adversely affect human fecundity; therefore, concerns have been raised regarding reproductive health consequences after exposure [6]. The consequences involve disorders in the functioning of the male and female reproductive systems, including birth defects, pre-term birth, developmental disorders, low birth weight, impotence, reduced fertility, and menstrual disorders [7]. Examples of these contaminants include Bisphenol A (BPA), present in polycarbonate plastics and used in a variety of common consumer products. It has been associated with lowered human sperm concentration, higher sperm velocity ratios and lower sperm swing characteristics [8]. Another is glyphosate, an active ingredient of the commonly used herbicide Roundup, which has been found to exert toxic and adverse effects on sperm progressive motility at 0.36 mg/L of Glyphosate and sperm motility at 1 mg/L of Roundup [9,10].

Exposure to heavy metals during pregnancy has been associated with adverse effects on the development of the gonads [4]. They may act as testicular toxicants and correspond to different compounds, which are related to social habits, life conditions, working hazards or use of drugs and medicines [11,12,13]. Metals are present in food, dietary supplements, water, air, alcoholic drinks, and tobacco [14]. Some heavy metals have demonstrated potent estrogenic and androgenic activities in vivo and in vitro by directly binding to steroid receptors, disrupting spermatogenesis and steroidogenesis, and resulting in decreased sperm concentration and motility [1,15]. They can affect the testis size, the semen quality, the secretory function of the prostate and seminal vesicles, the reproductive endocrine function and may lead to the loss of fertility and libido or to impotence [16,17,18,19,20]. Heavy metal exposures can also increase the levels of reactive oxygen species (ROS), oxidative stress, DNA damage, and disruption of the blood–testis barrier causing sperm apoptosis [21]. Lead (Pb), cadmium (Cd), and mercury (Hg) exert a negative impact on male reproductive health either by a direct effect on the target gland or indirect effects on the neuroendocrine system even at a low level of exposure [22]. Heavy metals such as manganese (Mn), copper (Cu), chromium (Cr), molybdenum (Mo), selenium (Se), and zinc (Zn) are important for maintaining good health by aiding as supplementing components for numerous enzymes; however, an accumulation to toxic concentrations may be harmful to humans [23,24].

Given that half of the men with fertility problems suffer from idiopathic male infertility [25] and the limited predictive value of basic semen analysis for pregnancy in couples trying to achieve natural conception, the need for more extended sperm functional testing has been highlighted [26,27]. Such in-depth assessment assists to identify sperm dysfunction at the cellular and molecular level [28] and supports more accurate prediction of the in vitro and in vivo fertilizing ability of human spermatozoa [29]. Several sperm functions are required by spermatozoa to reach and ultimately fertilise the oocyte, including sperm motility, viability and vitality, cervical mucous penetration, acrosome reaction, hyperactivation, and zona pellucida binding [30,31].

The current study has focused on two heavy metals recognised to have a negative effect on the male reproductive system, namely copper and cadmium. Cigarette smoke is one of the most important sources of cadmium exposure in the general non-occupationally exposed population [14]. Copper has been listed as a pesticide by the United States Environmental Protection Agency (USEPA, 2008) and its compounds are used extensively in various agricultural settings [32], indicating a great risk of exposure through skin absorption. In a previous study, we reported on the detrimental effects of copper sulphate (CuSO_4_) and cadmium chloride (CdCl_2_) on human sperm motility and vitality at concentrations of 250 μg/mL and 500 μg/mL, respectively [33]. In particular, a significant decrease was observed in the sperm velocity parameters, which are closely related to fertility prediction and pregnancy outcomes in humans [34] and animals [35,36]. Moreover, Marchiani et al. [37] indicated that human spermatozoa exposed to 10 µM CdCl_2_ had decreased progressive and hyperactivated motility and a higher percentage of induced sperm acrosome reaction. There are various reports on the effect of heavy metals on human sperm quantity and quality but similar studies on non-human primates (NHP) sperm structure and function are limited.

Non-human primates (NHPs) have been identified as key models in human-related studies and are often used in research on male fertility/infertility, in vitro fertilization (IVF) or assisted reproductive technology (ART) procedures [38], male contraception [39], and reproductive toxicology [40,41]. For an animal model to be pharmacologically relevant for developmental and reproductive toxicity (DART) testing, the test extract must produce a similar in vitro or in vivo pharmacokinetic effect as found in humans [42]. However, comparing the results of NHP and human studies require objective, standardised, and sensitive techniques to recognise compromised sperm function.

The purpose of our study was to evaluate the suitability and efficacy of sperm functional tests using NHP spermatozoa. In this regard, a concomitant aim was to evaluate the effect of CuSO_4_ and CdCl_2_ on selected sperm functions for three NHP species. It is envisaged that our results will assist to affirm the use of NHP models for reproductive toxicology studies.

## 2. Materials and Methods

The optimization and standardization of sperm functional tests for primate spermatozoa have been described by Prag (2017) [43].

### 2.1. Species Studied

The species of focus for this study were the vervet monkey *(Chlorocebus aethiops)*, the chacma baboon (*Papio ursinus*) and the rhesus monkey (*Macaca mulatta)*. All three species are members of the family Cercopithecidae and sub-classified as Old World primates. The vervet monkey is widely distributed in Africa and one of the few non-human primates that are successfully bred under controlled captive conditions [44] throughout the year; therefore, semen samples are readily available [45]. The only species of baboon studied in any detail and reported in the literature is the olive baboon (*Papio anubis*); however, the chacma baboon (*Papio ursinus*) is widely distributed in southern Africa [46]. Since baboons have adapted to diverse habitats, they have broken free of the seasonal constraints of their habitats, and therefore show virtually no seasonality in reproduction [47]. The rhesus macaque has been used extensively as a model for ART and early human development as well as for fertilization and embryonic development in primates [48]; however, they are seasonal breeders and higher quality samples are only available for approximately 4 months of the year [45].

All three primates species were housed at the Primate Unit and Delft Animal Centre of the South African Medical Research Council (Cape Town, South Africa) according to the husbandry guidelines of the South African National Standard for the Care and Use of Animals for Scientific Purposes (South African Bureau of Standards, SANS 10386).

Seven adult captive-bred male vervet monkeys, between the ages of 6 and 10 years [45] and a weight ranging from 4.4 to 7.4 kg, were selected from an indoor breeding colony, with a 30-year history of successful sixth-generation reproductive performance. The environmental conditions were maintained at temperatures between 25 and 27 °C, a humidity of 45%, 15-air changes/h and a photoperiod of 12 h. The monkeys were fed a soft diet consisting of pre-cooked maize enriched with a protein supplement, vitamins, minerals, trace elements, and supplemented with seasonal fruit. Drinking water was available ad libitum via an automated water system.

Six adult wild-caught male baboons, with an average age of 15 years and weight ranging from 24.1 to 28.7 kg, were housed outdoor in single cages, adjacent to a female with full visual, olfactory, and auditory contact. Grooming was possible between adjacent individuals through panels of wired mesh. Six captive-bred male rhesus monkeys ranged between the ages of 10–14 years and 6.7–12.3 kg in weight were selected from an outdoor breeding colony. The breeding stock is of Chinese origin and the colony has a fifteen-year history of successful second-generation reproductive performance. These males were temporarily housed indoors and each paired with a compatible breeding female in double galvanized steel cages. Their environmental conditions were maintained at temperatures ranging from 25 to 27 °C and 12 h light/dark controls. Both baboons and rhesus monkeys were fed a standard diet of pelleted feed (Aquafeeds, Cape Town, South Africa) and seasonal fruit or vegetables. The diet was supplemented with bread slices covered with vitamin C syrup (Portfolio Pharmaceuticals, Johannesburg, South Africa). In addition, the animals were provided with foraging logs three times per week and various other enrichment devices. Drinking water was available ad libitum via an automated water system.

### 2.2. Sample Collection and Evaluation

Ethical clearance for this study was obtained from the Ethics Committee of the University of the Western Cape (Ethical clearance number: 13/10/91) as well as the Ethics Committee for Research on Animals of the South African Medical Research Council (Project number: 11/13). All the animals remained in good health and none of these males had been used in other experimental procedures.

Samples were collected twice a week, including four males per week (two males per day). Sampling took place in the early morning, due to the males’ high incidence of masturbation, and males were separated from the females a day or 12 h before to prevent copulation. Semen collection was performed under anaesthesia (ketamine hydrochloride (InterVet, Johannesburg, South Africa) at a dose of 10–15 mg/kg by intramuscular injection) via rectal probe electro-ejaculation, a method described by Seier et al. [49] and Cseh et al. [50], into sterile pre-warmed 15 mL or 50 mL plastic containers. The samples were incubated in a temperature-controlled oven with a constant temperature at 37 °C for 5 min after collection to allow the semen coagulum to liquefy. Twenty semen samples were selected from each species based on the semen quantity and quality (volume > 350 µL, sperm concentration > 20 × 10^6^ mL and percentage total motility > 50%) for further experimentation.

Due to the limited availability of good quality semen samples, certain functional tests could not be performed for the baboon (WST-1 assay and hyperactivation) and the rhesus monkey (vitality and acrosome integrity), respectively. These limitations were due to, as previously mentioned, the high incidence of male masturbation in the mornings as well as seasonality of breeding.

### 2.3. Selection of Motile Spermatozoa

Human tubal fluid (HTF), supplemented with 1% human serum albumin (HSA), was used for all experimental procedures [51]. All chemicals were supplied by Sigma Aldrich (Cape Town, South Africa).

A double-density gradient centrifugation technique was employed to select motile spermatozoa by using PureSperm (Kat Medical Laboratories, Johannesburg, South Africa). This procedure involved layering 50–300 µL PureSperm 80 into a 1.7 mL conical centrifuge tube, followed by equal volumes of PureSperm 40 and liquefied semen. The preparation was centrifuged at 300× *g* for 20 min, followed by removal of the supernatant. The sperm pellet was then washed by resuspending it in 100 µl HTF (without HSA) and centrifugation at 500× *g* for 10 min, followed by removal of the supernatant. The remaining pellet was resuspended in 100 µL HTF (with HSA), with or without a specific metal concentration, to result in a final concentration of 10–20 × 10^6^ mL motile spermatozoa. Sperm preparations were kept at 37 °C before sperm functional tests were performed.

### 2.4. Exposure of Spermatozoa to Heavy Metals

Sperm preparations were exposed to different concentrations of CuSO_4_ (10, 50, 100, and 250 µg/mL) and CdCl_2_ (10, 50, 100, and 500 µg/mL) as reported in a previous study on human spermatozoa [33]. All metal concentrations were prepared using HTF supplemented with 1% HSA. Motile sperm preparations were exposed to these metal concentrations for up to 3 h while being incubated at 37 °C and sperm functionality were assessed at various time intervals as indicated in the tables and figures. All sperm motility assessments were done in triplicate (n = 3) with three different males’ samples used for each experiment. Sperm viability and vitality, acrosome intactness and hyperactivation were assessed using 3–7 individual semen samples for each test.

### 2.5. Sperm Structural and Functional Testing

#### 2.5.1. Motility Analysis

All sperm parameters and sperm motion characteristics were evaluated using the Motility module of the Sperm Class Analyzer^®^ (SCA) (Microptic S.L., Barcelona, Spain) computer-aided sperm analysis (CASA) system, Version 5.1. Capturing of the data involved a Basler A602fc, A312fc or aVA 1000–100gc digital camera (Microptic S.L., Barcelona, Spain) that was mounted (C-mount) onto a Nikon Eclipse 50i microscope (IMP, Cape Town, South Africa) or an Olympus CH2 microscope (Wirsam, Cape Town, South Africa), both equipped with a 10× phase contrast objective and a heated stage.

Sperm motility analysis involved pipetting 2 µL of sperm preparation into a 20 µm deep pre-warmed (37 °C) 8 chamber Leja slide (Leja Products B.V., Nieuw-Vennep, The Netherlands) before it was assessed. Various motility parameters were determined such as percentages total motility, progressive motility, and non-progressive motility, as well as rapid, medium, and slow swimming spermatozoa. Eight kinematic parameters were calculated by capturing tracks at 50 frames/s until a total of 200 motile spermatozoa were analysed. The kinematic parameters included “Velocity” (curvilinear velocity (VCL), straight-line velocity (VSL) and average path velocity (VAP)), “Linearity” (linearity (LIN = VSL/VCL) and straightness (STR = VSL/VAP)) and “Vigour” (wobble (WOB = VAP/VCL), amplitude of lateral head displacement (ALH) and beat-cross frequency (BCF)). The ALH parameter was measured as half the width of the VCL track and not as the full VCL wave or doubling of the riser values (risers method) as described by Mortimer [51,52]. The three sperm subpopulations were determined using VCL cut-off values previously established [53] of 50 < 80 > 120 µm/s to identify slow, medium, and rapid swimming spermatozoa. Fields were captured randomly to avoid bias toward higher sperm motility; however, fields containing debris or clumps of spermatozoa were avoided to limit incorrect analysis.

#### 2.5.2. Viability and Vitality Analysis

##### Eosin-Nigrosin (E-N)

Thirty microliters of preheated eosin-nigrosin staining solution were placed on a coverslip on a heated stage (37 °C) whereafter 10 µL sperm preparation was mixed with the solution for 30 s. From this mixture, smears were made using 10 µl aliquots and left to air-dry overnight before being mounted with a coverslip and DPX (Sigma, Cape Town, South Africa). Spermatozoa were assessed using a Nikon Eclipse 50i microscope (IMP, Cape Town, South Africa) and a 40× objective. At least 100 spermatozoa per samples were assessed for viability and the percentage live (white) and dead (pink) spermatozoa were calculated.

##### Hoechst and Propidium Iodide (H&PI)

Staining solution aliquots (1 mg/mL) of Hoechst 33342 (trihydrochloride trihydrate) and propidium iodide (PI, Sigma, Cape Town, South Africa) were pre-heated (37 °C). Ten microlitres of sperm preparation were first added to 1 µL Hoechst and incubated for 5 min at 37 °C. Thereafter, 1 µL PI was added to the mixture and incubated for another 5 min. After incubation, 10 µL of the mixture was placed on a slide with a coverslip and viewed under a Nikon Eclipse 50i fluorescence microscope (IMP, Cape Town, South Africa), equipped with a 20× objective and a DAPI filter. At least 100 sperm were captured with NIS Elements imaging software (IMP, Cape Town, South Africa) and manually assessed for viability to calculate the percentage live (blue) and dead (red) spermatozoa.

##### WST-1 Cell Proliferation Reagent

WST-1(sodium 5-(2,4-disulfophenyl)-2-(4-iodophenyl)-3-(4-nitrophenyl)-2Htetrazoliuminner salt; Cell Proliferation Agent; Cat No. 11 644 807 001) is commonly used in tissue culture studies for cell proliferation, vitality, and cytotoxicity assessments (Roche, Mannheim, Germany) and has proven to be sensitive to changes in sperm vitality [33]. It is a ready to use solution, containing WST-1 and an electron coupling reagent, diluted in phosphate-buffered saline (PBS) [54]. This test is based on the principle that the tretrazolium salt WST-1 is cleaved to formazan by cellular enzymes or reduced at the external surface of the plasma membrane by NADH and NADPH oxidase [55]. An expansion in the number of viable cells results in an increase in the overall activity of mitochodrial dehydrogenases in the sample. An augmentation in enzyme activity leads to a increase in formazoan dye which correlates to the number of metabolically active cells, indicating vitality.

The WST-1 labeling reagent was prepared according to the manufacturer’s protocol by mixing 200 µL WST-1 reagent with 800 µL PBS. Equal volumes (50 µL) of sperm preparation and labeling reagent were added to the wells of a microtitre plate in duplicate for each metal concentration. A zero min reading was obtained where after plates were incubated at 37 °C and 5% CO_2_ until the next analysis. Measurements of the absorbance were carried out after 1, 2, and 3 h of incubation using an ELISA reader (Microplate reader, Multiskan EX, Thermo Scientific, Johannesburg, South Africa) set at 450 nm with a reference wavelength of 650 nm. Using these absorbance measurements, IC_50_ values, defined as the concentrations of the heavy metals required for 50% inhibition of sperm vitality, were determined as a parameter for the toxicity of CuSO_4_ and CdCl_2_ to NHP spermatozoa.

##### Evaluation of Acrosome Intactness

Fluorescein-conjugated Pisum sativum agglutinin (FITC-PSA) (Sigma, Cape Town, South Africa) was used to determine the acrosomal intactness of the spermatozoa. FITC-PSA working solution was prepared by adding 50 µL FITC-PSA to 450 µL PBS and was kept at 4 °C. Sperm preparations were diluted with Ham’s F10 (Invitrogen, Thermo Fisher Scientific, Johannesburg, South Africa) to yield a sperm concentration of 2 × 10^6^ mL. Two 5 µL drops of the diluted sperm sample were placed on a slide, spread in a circle and left to air dry overnight. Dried smears were fixed with 95% ethanol for 30 min at 4 °C and left to air dry for a further 30 min. Five microlitres of FITC-PSA solution was added to each smear and kept in the dark for 45 min. The slides were then washed in dH_2_O for a few seconds and left to dry upright, whereafter they were mounted with coverslips using Dako fluorescent mounting medium (Diagnostech, Johannesburg, South Africa). The slides were analysed with a Nikon Eclipse 50i fluorescence microscope (IMP, Cape Town, South Africa) using a green fluorescence filter and 100× objective. At least 100 spermatozoa were captured with the NIS Elements software and manually assessed to calculate the percentage of intact acrosomes (bright green) and reacted (dark or with a centre band) acrosomes.

##### Evaluation of Hyperactivation

Caffeine was used to induce sperm hyperactivation in vervet and rhesus monkey samples, similar to a previous study on human spermatozoa [56]. The effect of CuSO_4_ and CdCl_2_ on sperm hyperactivation was evaluated by exposing spermatozoa to 100 µg/mL of both heavy metals respectively. Using a positive displacement pipette, 1 µL of semen was placed into individual chambers of a pre-heated 4 chamber Leja slide. Following the flush technique [57], 2 µL of caffeine (5 mM), CuSO_4_, CdCl_2_ or a mixture of the caffeine and heavy metal suspension was used to flush the semen into the Leja slide chambers, whilst keeping the slide on a heated stage. Sperm motility was immediately evaluated (see Section 2.5.1) and every 15 min thereafter for an h or until motility decreased below the WHO [58] reference limits (motility > 40%). The kinematic parameter cut-off values employed for identifying hyperactivated sperm for the rhesus monkey, were adapted to VCL ≥ 130 µm/s, LIN ≤ 69%, and ALH ≥ 7.5 µm (≥3.75 µm), as previously described by Baumber and Meyers [59].

##### Statistical Analysis

The MedCalc^®^-software 12.3.0 (Mariakerke, Belgium) was used for statistical analysis. One-way analysis of variance analysis (ANOVA) was performed for parametric data distributions. Any significant differences indicated in the ANOVA table between groups were analysed further using the Student–Newman–Keuls test for pairwise comparisons. The Kruskal–Wallis test was employed for non-parametric data distributions and further elaborated for individual differences using the Mann–Whitney test for independent samples. Data are presented as the mean percentage ± standard deviation (SD) in the tables and *p* < 0.05 was considered significant.

## 3. Results

### 3.1. Motility

Sperm motility parameters presented a tendency to decrease over time (Appendix A) and indicated the negative effect of the higher metal concentrations (50, 100, 250, or 500 µg/mL). In all three primate species, a significant decrease (*p* < 0.05)was found in a combination of percentage sperm motility (total, progressive, rapid-, and medium-swimming) (Appendix A) and sperm velocity (VCL, VSL, and VAP) as well as the linearity (LIN and STR) and vigour (WOB, ALH, and BCF) of swimming tracks (Appendix A) after exposure to CuSO_4_ and CdCl_2_.

Examples of the effect of these two metals on percentage motility, swimming speed and linearity of sperm tracks of individual males are displayed in Figure 1, Figure 2 and Figure 3. In both the vervet and rhesus monkey, these three motility parameters presented an onset of significant data at the highest metal concentrations from the 15 min time point and also at the 50 and/or 100 µg/mL concentrations from the 60 min time point (Figure 1 and Figure 2). In the baboon, the most significant effects on sperm motility were found from the 45 min time point for the highest metal concentrations (Figure 3).

### 3.2. Viability and Vitality

#### 3.2.1. Eosin-Nigrosin (E-N) and Hoechst and Propidium Iodide (H&PI) Staining

After CuSO_4_ and CdCl_2_ exposure, there was no significant effect on viability with E-N staining for vervet spermatozoa. After 120 min of exposure to 250 µg/mL CuSO_4_, the percentage live spermatozoa were 28.4% compared to the control (27.4%) and for 500 µg/mL CdCl_2_, the percentage was 31.8% compared to the control (32.4%) (data not shown). However, with H&PI staining the percentage live vervet monkey spermatozoa was significantly decreased at 100 µg/mL and 250 µg/mL CuSO_4_ (*p* = 0.018) and 500 µg/mL CdCl_2_ (*p* = 0.001) after 120 min of exposure (Table 1). For baboon spermatozoa with E-N staining, exposure to CuSO_4_ revealed no significant effect for both time points. Exposure to CdCl_2_ revealed a significant decrease in the percentage live baboon spermatozoa after 15 min at the highest concentration, 500 µg/mL (*p* = 0.043). After 90 min of exposure, a general trend was noted for a decrease in percentage live spermatozoa with an increase in CdCl_2_ concentration (Table 1), albeit no significance was found, possibly due to large standard deviations. These large standard deviations are due to the heterogeneity of primate spermatozoa causing individual differences in semen quality for each sample and/or ejaculate in response to the metals. For H&PI staining, only one sample was evaluated for baboon spermatozoa and a potential decrease in percentage live spermatozoa was noted after exposure to the highest concentrations of CuSO_4_ and CdCl_2_ after 60 and 75 min (data not shown).

#### 3.2.2. Vervet WST-1 Cytotoxicity Assay

The WST-1 cytotoxicity assays for vervet monkey spermatozoa upon exposure to both metals revealed significant decreases in absorbance readings after 180 min at most concentrations of the metals (Appendix A). These absorbance readings were used to calculate the IC_50_ values for CuSO_4_, namely, 21.2 µg/mL, while the IC_50_ value for CdCl_2_ could not be determined. However, due to irregular absorbance readings brought on by the heterogeneity of the data, the results reported here should only be seen as an estimate of the effect of heavy metals on NHP sperm vitality.

### 3.3. Acrosome Intactness

Vervet monkey sperm acrosomes were not affected by exposure to the two metals up to 120 min exposure time points (Table 2). Baboon spermatozoa revealed a significant decrease in percentage intact acrosomes (*p* = 0.005) after 15 min of exposure to 250 µg/mL CuSO_4_. A similar significant decrease was found after 90 min (*p* < 0.001) at 50, 100, and 250 µg/mL CuSO_4_. Interestingly, no effect was found on the percentage intact acrosomes for baboon sperm at the different CdCl_2_ concentrations for both time points (Table 2).

### 3.4. Hyperactivation

A definite trend was noted for the effect of CuSO_4_ with the lowest percentage hyperactivation recorded when spermatozoa were exposed to 100 µg/mL CuSO_4_ at each time point (Table 3); however, a significant decrease was only found at 40 min of exposure (*p* = 0.012). When vervet monkey spermatozoa were exposed to 100 µg/mL CdCl_2_, a significantly lower percentage hyperactivation (*p* = 0.007) was recorded at each time point compared to the control, caffeine, or metal–caffeine combination.

Although 5 mM caffeine does not seem to induce hyperactivation in vervet monkey spermatozoa, the metal–caffeine combination maintained the percentage hyperactivation compared to the control. For the rhesus monkey spermatozoa, only one sample was evaluated for each metal and thus the metals’ true effect could not be deduced, but a definite effect was seen. Exposure to the metals revealed a similar effect and after 50 min of exposure, the 100 µg/mL metal concentrations caused a decrease in the percentage hyperactivation of 43–78%, while the metal-caffeine combination maintained the values to equal or slightly higher than the control (Table 3).

## 4. Discussion

During the past four decades, living populations have been exposed to rising levels of environmental contaminants which ultimately accumulate in the organisms and induce multiple organ alterations due to its toxicity [37]. These compounds are present in the environment (air, water, and soil), mass-consumer products and food products where it may operate as endocrine disruptors. Subsequently, environmental (bisphenol A and parabens), nutritional (diet, and body mass index (BMI)) and lifestyle factors (physical activity, and smokers), could all be risk factors linked to increased incidence of male infertility and the decline of human semen quality [60,61].

Although some persistent contaminants have been quantified in seminal fluids, limited information has been reported on what these chemical concentrations may mean for reproductive function. Studies have been evaluating a select number of chemicals with basic semen analysis that focus on sperm count, motility, and morphology despite modern methods to evaluate additional functional measures that have been related to fecundity [6]. This calls for more investigation into alternative methods for functional testing, the use of non-human primates as research models and toxicity tests. Similar to findings in humans [33], the two heavy metals investigated in our study, namely, copper sulphate (CuSO_4_) and cadmium chloride (CdCl_2_), indicated a detrimental effect at high doses on NHP sperm function and may possibly contribute to impaired sperm fertilising capabilities and idiopathic male infertility.

### 4.1. The Effect of Heavy Metals on Sperm Motility

Motility is one of the most important characteristics associated with the fertilising ability of spermatozoa; thus, the examination of this parameter constitutes an integral part of semen analysis [62]. Exposure of NHP spermatozoa to various metal concentrations presented a decrease in sperm motility parameters (percentages in sperm subpopulations as well as velocity, linearity, and vigour of swimming tracks) over time for the higher concentrations of both CuSO_4_ and CdCl_2_. Although a slight stimulatory effect is noted at the 60 and 45 min time point for the lower concentrations, this might be an indication of the phenomenon of hormesis, where low doses of an agent may be beneficial, but are however, toxic at high doses. Therefore, it is clear that high concentrations of these metals had negative effects on sperm motility parameters. A decline in control samples was also evident and expected, as sperm motility naturally declines over time; however, even though this is a noticeable change, the spermatozoa exposed to the heavy metals always presented a lower motility than the control over time. For human spermatozoa, a significant decrease was reported in percentage total and progressive motility after 24 h of exposure to 10 µM CdCl_2_ as well as VAP, VCL and STR. Similarly, Hardneck et al. [33] also found a significant negative effect on sperm motility at the highest concentrations of these two metals (250 μg/mL and 500 μg/mL). In turkey spermatozoa, it was reported that after 30 min of exposure, CuSO_4_ (12.5 μg/mL to 50 μg/mL) significantly lowered the percentage total and progressive motility, as well as the values for VCL, ALH, and BCF [63].

Metals may impair progressive sperm motility by accumulating in the epididymis, prostate, vesicular seminalis, or seminal fluid [64]. More so, the potential for chemicals to cross the placental barrier [65,66] in trace concentrations may result in serious damage in new-borns and raises great concern. Copper toxicity leads to ROS production followed by protein and lipid oxidation [67], which negatively correlate with sperm motility and viability [68]. Once the copper is taken up into a cell, the excess of it is reduced to cuprous ions that readily bind with sulfhydryl groups [69], interfering with electron transport and inhibiting ATP production [70]. Copper also accumulates in the sperm mitochondria [71], decreasing the mitochondrial membrane potential while causing ROS formation and resulting in oxidative damage [72]. The action of cadmium chloride may be explained by its effects on microtubules and sperm mitochondrial function [73]. Previous studies have indicated that cadmium inhibits microtubule sliding and affects sperm mitochondrial function [73,74]. Cadmium also competes with calcium for calmodulin binding, and therefore this inhibition results in a decrease in sperm motility [75,76]. Albeit all the aforementioned studies investigated the in vitro effect of these metals on sperm motility, it is conceivable that if a male is exposed to high concentrations of the metals, it could be detrimental to the fertilization potential of sperm. Therefore, the objective measurement and quantification of sperm motility parameters, using CASA instruments, are preferred, rather than subjective manual sperm motility assessments [77,78,79].

### 4.2. The Effect of Heavy Metals on Sperm Viability and Vitality

Viability and vitality represent two different aspects of cell functions, and both are required for the estimation of the physiological state of a cell after exposure to various types of stressors and factors [80]. While viability defines the percentage of live cells in the whole population, vitality defines the physiological capabilities of a single cell. The toxic effects of chemicals or physical factors do not necessarily lead to cell death, but may cause several morphological, intracellular, or metabolic alterations [80], which are not defined by viability tests such as staining techniques. These metabolic alterations may therefore be determined through vitality tests such as cytotoxicity assays.

Previous studies have indicated a significant decrease in sperm viability after exposure to CuSO_4_ [81] and CdCl_2_ [82] when utilizing the E-N stain on mammalian spermatozoa; however, this was not always the case in the current study. A significant decrease in the percentage of live vervet spermatozoa, at higher concentrations of both heavy metals, was observed after 120 min of exposure as indicated by the H&P stain. In addition, only CdCl_2_ exposure caused a significant decrease in the percentage live baboon spermatozoa as indicated by E-N staining. The use of the E-N stain to test NHP spermatozoa viability, after exposure to the two heavy metals, thus requires an extended incubation period to validate its sensitivity, in particularly to CuSO_4_. The reason for this anomaly is not clear; the possibility that the technique used is not compatible with NHP spermatozoa can be ruled out due to previous studies on spermatozoa from drill (African forest baboon) [83] and rhesus monkey [84] successfully employing E-N viability testing of primate spermatozoa.

The H&PI stain revealed a significant effect only at the last time point for both metals, suggesting an extended incubation period, over 120 min, for the test to detect a more noticeable change in vervet sperm viability. The E-N baboon results indicated that only the 500 μg/mL CdCl_2_ affected viability with significant changes observed from the first time point of testing. In a *Drosophila* study, the LIVE/DEAD assay (SYBR^®^ and Propidium iodide) was used to measure sperm viability after exposure to CdCl_2_ (10 μM to 500 μM CdCl_2_) and reported an increase in the number of dead sperm proportional to increasing amounts of CdCl_2_ [85]. In rats, the possible mechanism behind this effect was reportedly DNA fragmentation after CdCl_2_ exposure [73]. Although DNA fragmentation was not assessed in our study, these previous findings support the effects of copper and cadmium seen on sperm viability for NHPs.

The WST-1 assay proved to be successful in demonstrating the negative effects of both metals on vervet sperm vitality. According to Roche [54], the reagent (tetrazolium salt) is cleaved by enzymes (mitochondrial dehydrogenase) to formazan. The higher number of viable spermatozoa results in higher dehydrogenase activity which in turn increases the amount of formazan formed; this directly correlates to the number of metabolically active spermatozoa [54]. Significant decreases in absorbance were found for most CuSO_4_ concentrations and all concentrations of CdCl_2_ after 180 min of exposure. These results are in accordance with a previous study investigating human sperm vitality, during which the WST-1 assay indicated the direct proportion of vital cells [33]. This correlates with our motility results, as a significant decrease was observed over time for both metals after exposure to the higher concentrations for all three primates (see Section 3.1). Furthermore, in order to determine the exact concentration at which the metals exert their effect, we calculated the IC_50_ values to obtain the cut-off concentrations. Taking into consideration the WST-1 readings determined from the vervet spermatozoa, the calculated average IC_50_ value for CuSO_4_ was 21.2 µg/mL, which is less than the 50.3 µg/mL determined for human spermatoza [33]. Since 30–200 µg/mL copper in water has moderate to severe effects on human health (ranging from gastrointestinal irritation, nausea, and vomiting) and >200 µg/mL leading to poisoning with possible fatalities [86], these calculated IC_50_ values for both vervet and human sperm indicate that similar copper concentrations will also affect sperm function. The lower value for vervet spermatozoa might be an indication that NHP spermatozoa are more sensitive to CuSO_4_ but needs to be validated due to the general heterogeneity of NHP samples in this study. Although the aforementioned studies validate our results for the vervet samples, future investigations are required to prove the sensitivity of WST-1 for other NHP species.

Aitken et al. [87], indicated that human spermatozoa can chemically reduce the WST-1 reagent and this significant effect was also seen after 180 min of incubation. A further study by Knazicka et al. [88] found sperm vitality to be decreased by all concentrations of copper (3.9 μM/L to 1000 μM/L) via MTT (metabolic activity) assay and reported that high doses of copper are toxic to sperm motility. In a study by Dawson [89], the effect of cadmium in seminal plasma, was compared to sperm viability and it was found that the metals’ levels were inversely correlated with the percentage of live sperm, (cadmium, *p* < 0.01). Previous studies also found that cigarette smoking (of which cadmium is a component) has detrimental effects on sperm vitality [90]. This cadmium-induced toxicity effect is believed to be due to increased oxidative stress [91], the generation of ROS resulting in oxidative deterioration of lipids, proteins, and DNA [92], which impairs sperm quality.

### 4.3. The Effect of Heavy Metals on Acrosome Intactness

The acrosome cap is part of the sperm head and surrounds the upper 40 to 60% of the sperm nucleus [93]. Mammalian fertilization requires sperm capacitation and acrosome reaction; the latter process being a major event before spermatozoa can penetrate the zona pellucida and subsequently fuse with the egg membrane [94].

Vervet monkey and baboon sperm acrosomes were evaluated for intactness; however, only baboon spermatozoa were found to present with a significant decrease in percentage intact acrosomes after exposure to CuSO_4_. Although our results indicated that CdCl_2_ had no significant effect on both vervet monkey and baboon spermatozoa, the test did detect a difference between intact and reacted acrosomes. It is however possible that either metal did not affect the acrosome intactness of the spermatozoa, or the reagent inaccurately stained the acrosomal contents.

A rabbit study by Roychoudhury et al. [95] reported a higher occurrence of spermatozoa with disordered membranes after copper exposure (3.57 to 4.85 μg CuSO_4_/mL), which proposed alterations in the anterior part of the head (acrosome). These findings support the decrease in intact acrosomes observed in the baboon samples, suggesting that the metal affected the acrosomal membrane resulting in the release of the acrosomal content. Misro et al. [96], reported that a high release of copper drastically lowers human sperm motility and viability, but only marginally affected the acrosome status. This supports the insignificant results obtained when evaluating the vervet samples and indicates possible species differences in the effect of this heavy metal on sperm acrosome intactness.

### 4.4. The Effect of Heavy Metals on Sperm Hyperactivation

Hyperactivation is characterised by the development of asymmetrical, high amplitude flagellar beats, causing vigorous and sometimes the non-directed movement of free-swimming spermatozoa [97]. The flush technique [56] was implemented, where spermatozoa would subsequently swim into the hyperactivation medium with the technique showing a high positive correlation with routine swim-up preparations and appeared to accelerate the process of hyperactivation [62]. Vervet spermatozoa were found to have a significantly decreased percentage hyperactivation after exposure to CuSO_4_ at the last time point (40 min). In contrast, the same concentration of CdCl_2_ caused a significant decrease at each time point (15, 20, and 40 min) for all three experimental groups.

When rhesus monkey spermatozoa were exposed to environmental tobacco smoke (of which cadmium is a major component [98]) it resulted in decreased percentages sperm motility and hyperactivation after incubation in cAMP and caffeine [99]. These results agree with the current study where the vervet samples were affected, as well as the lower values obtained for the rhesus sample.

## 5. Conclusions

The purpose of the current study was to evaluate the suitability of three NHP species as research models for sperm functional testing. This would allow comparisons of human and NHP sperm function after exposure to environmental contaminants, which may reveal or explain the high infertility rates in humans. While the vervet monkey samples were successfully used for all testing, providing results for analysis performed over time, all three NHP species could be used for sperm functional testing when acquired during breeding season. The two heavy metals, CuSO_4_ and CdCl_2_, provided substantial detrimental effects on sperm function by inhibiting sperm motility, viability, and vitality (observed via H&PI staining and the WST-1 assay), acrosome intactness and hyperactivation. Albeit these results prove that NHPs would be suitable models for toxicity testing and reproductive studies, future studies should include a larger sample size from all three species. Periods of analysis should be lengthened to further investigate whether the metals need time to elicit a more prominent effect for some functional tests. The evaluation of additional sperm functional tests and their sensitivity could also contribute to the assessment of NHP sperm function and should include the evaluation of sperm morphology, location and viability of sperm mitochondria, cervical-mucus penetration, zona pellucida binding, DNA fragmentation, and acrosome reaction induction.

## Figures and Tables

**Figure 1 ijerph-18-06200-f001:**
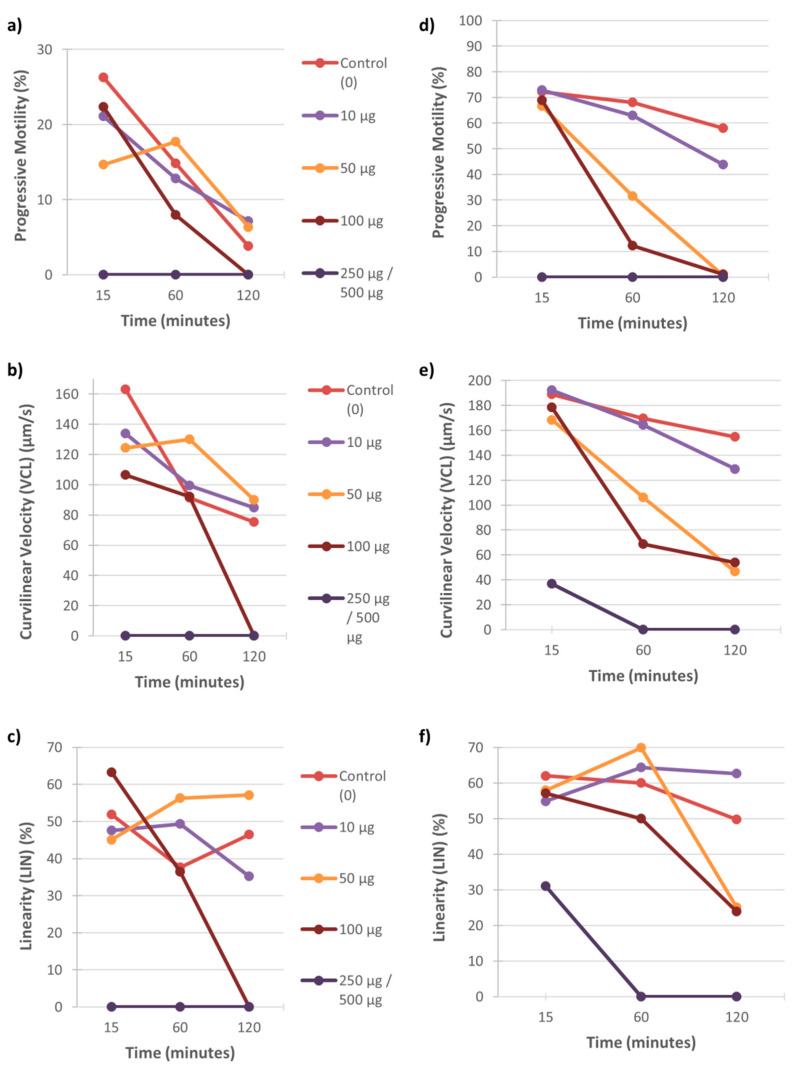
Vervet sperm motility parameters (progressive motility, VCL and LIN) over 120 min of exposure to five concentrations of CuSO_4_ (0–250 µg/mL) (**a**–**c**) and CdCl_2_ (0–500 µg/mL) (**d**–**f**). Each graph represents the data for one male to these metal concentrations.

**Figure 2 ijerph-18-06200-f002:**
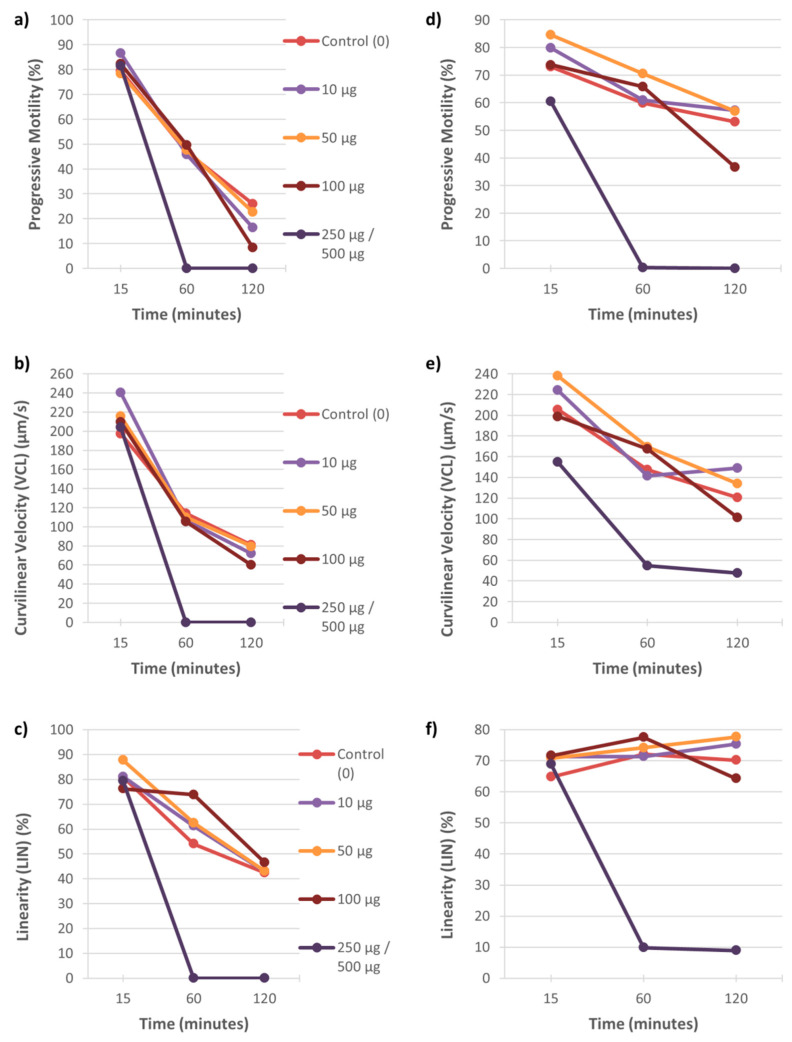
Rhesus sperm motility parameters (progressive motility, VCL and LIN) over 120 min of exposure to five concentrations of CuSO_4_ (0–250 µg/mL) (**a**–**c**) and CdCl_2_ (0–500 µg/mL) (**d**–**f**). Each graph represents the data for one male to these metal concentrations.

**Figure 3 ijerph-18-06200-f003:**
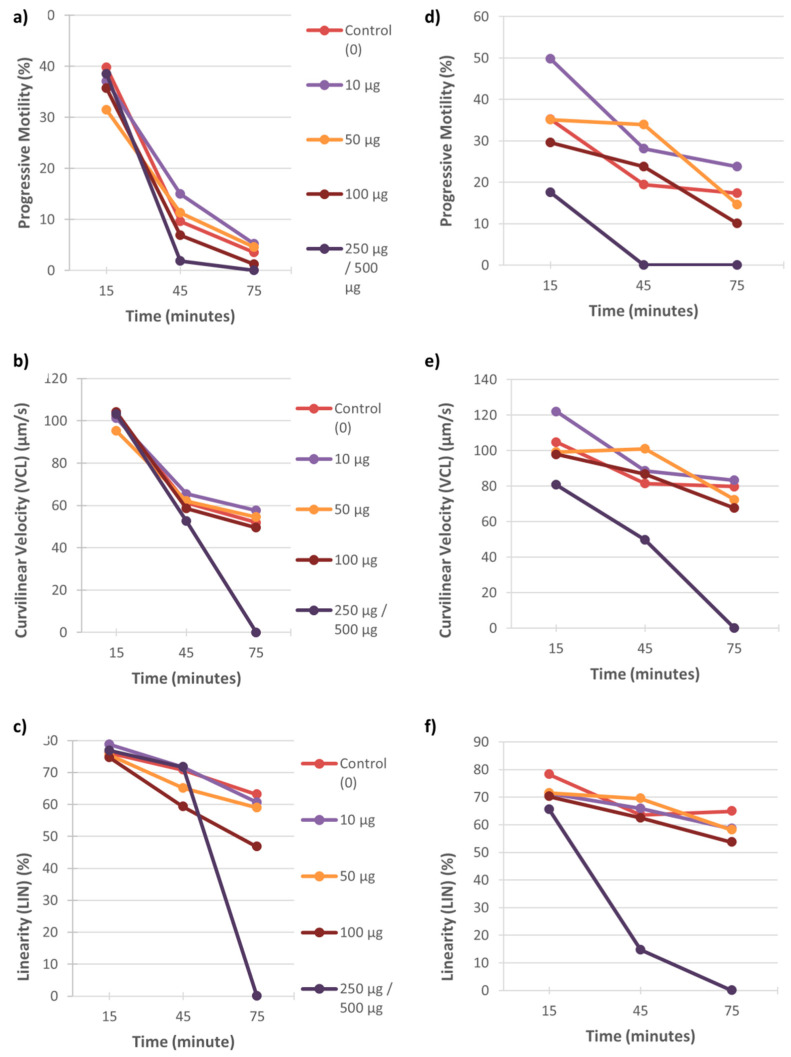
Baboon sperm motility parameters (progressive motility, VCL and LIN) over 75 min of exposure to five concentrations of CuSO_4_ (0–250 µg/mL) (**a**–**c**) and CdCl_2_ (0–500 µg/mL) (**d**–**f**). Each graph represents the data for one male to these metal concentrations.

**Table 1 ijerph-18-06200-t001:** Effect of CuSO_4_ and CdCl_2_ on sperm viability percentages (mean ± SD) of vervet monkey (n = 5) and baboon (n = 3) after 15–120 min incubation.

Time	CuSO_4_	Live Spermatozoa		CdCl_2_	Live Spermatozoa	
(min)	(µg/mL)	(%)		(µg/mL)	(%)	
		Vervet *^#^*	Baboon ^§^		Vervet ^#^	Baboon ^§^
15	0	54.0 ± 37.1	62.9 ± 11.4	0	53.2 ± 30.7	45.0 ± 39.5 ^a^
	10	54.5 ± 33.9	46.8 ± 13.5	10	49.3 ± 29.8	43.6 ± 10.3 ^a^
	50	48.6 ± 37.5	57.9 ± 8.8	50	57.5 ± 24.5	39.6 ± 11.3 ^a^
	100	33.2 ± 33.3	34.3 ± 14.8	100	41.4 ± 29.4	36.1 ± 15.3 ^a^
	250	19.3 ± 29.8	51.5 ± 18.4	500	46.6 ± 38.3	12.5 ± 12.0 ^b^
90/120 *	0	26.1 ± 28.3 ^a^	51.5 ± 20.6	0	50.9 ± 36.5 ^a^	33.8 ± 31.3
	10	30.0 ± 40.4 ^a^	55.7 ± 15.1	10	40.9 ± 28.8 ^a^	27.1 ± 13.8
	50	22.9 ± 39.6 ^a^	40.4 ± 7.5	50	38.4 ± 38.8 ^a^	28.7 ± 25.1
	100	0.0 ± 0.0 ^b^	31.9 ± 9.7	100	37.6 ± 36.1 ^a^	19.9 ± 24.1
	250	0.0 ± 0.0 ^b^	55.7 ± 17.1	500	0.2 ± 0.5 ^b^	9.8 ± 9.1

* = 90 min for baboon, 120 min for vervet; ^#^ = H&PI staining; ^§^ = E-N staining. ^a, b^ = values in columns with different superscript letters were significantly different among the concentrations of CuSO_4_ and CdCl_2_ (*p* < 0.05).

**Table 2 ijerph-18-06200-t002:** Effect of CuSO_4_ and CdCl_2_ on intact acrosome percentages (mean ± SD) of vervet monkey (n = 7) and baboon (n = 6) after 15–120 min incubation.

Time	CuSO_4_	Intact Acrosome		CdCl_2_	Intact Acrosome	
(min)	(µg/mL)	(%)		(µg/mL)	(%)	
		Vervet	Baboon		Vervet	Baboon
15	0	26.2 ± 22.9	74.3 ± 18.1 ^a^	0	43.5 ± 35.0	94.1 ± 7.2
	10	35.6 ± 24.8	77.9 ± 11.2 ^a^	10	54.1 ± 40.8	91.1 ± 10.4
	50	21.1 ± 26.9	60.2 ± 31.9 ^a^	50	52.1 ± 39.4	90.6 ± 12.5
	100	28.2 ± 33.3	55.6 ± 31.2 ^a^	100	47.2 ± 36.5	92.6 ± 8.8
	250	24.4 ± 28.3	19.2 ± 29.8 ^b^	500	48.8 ± 37.6	84.4 ± 7.5
90/120 *	0	34.4 ± 20.4	71.7 ± 13.2 ^a^	0	37.8 ± 36.1	91.5 ± 4.7
	10	34.2 ± 26.6	75.0 ± 12.8 ^a^	10	47.5 ± 35.5	90.9 ± 8.9
	50	27.8 ± 30.8	69.7 ± 17.5 ^b^	50	41.1 ± 33.1	90.2 ± 5.6
	100	28.2 ± 24.7	45.4 ± 23.0 ^c^	100	30.9 ± 35.8	91.6 ± 7.4
	250	25.0 ± 35.0	17.3 ± 27.0 ^d^	500	28.9 ± 23.2	85.2 ± 5.6

* = 90 min for baboon, 120 min for vervet. ^a, b, c, d^ = values in columns with different superscript letters were significantly different among the concentrations of CuSO_4_ (*p* < 0.05).

**Table 3 ijerph-18-06200-t003:** Effect of CuSO_4_ (100 μg/mL), CdCl_2_ (100 μg/mL) and caffeine (5 mM) on percentage sperm hyperactivation (mean ± SD) of vervet (n = 5) and rhesus monkey (n = 1) after 15–50 min incubation.

Time	CuSO_4_	Caffeine	Hyperactivation		CdCl_2_	Caffeine	Hyperactivation	
(min)	(µg/mL)	(mM)	(%)		(µg/mL)	(mM)	(%)	
			Vervet	Rhesus			Vervet	Rhesus
	0	0	10.1 ± 10.6	63.5	0	0	6.0 ± 1.2 ^a^	41.1
15 (V)	0	5	12.0 ± 3.2	70.4	0	5	14.1 ± 12.8 ^b^	71.2
10 (R)	100	0	4.2 ± 3.6	71.9	100	0	0.4 ± 0.8 ^c^	37.8
	100	5	16.8 ± 1.9	71.1	100	5	12.8 ± 18.5 ^d^	61.5
	0	0	7.6 ± 7.7	59.1	0	0	7.7 ± 6.6 ^a^	17.7
20 (V)	0	5	7.0 ± 7.1	68.8	0	5	11.2 ± 16.3 ^a^	82.3
35 (R)	100	0	3.4 ± 5.1	35.6	100	0	0.8 ± 1.7 ^b^	26.4
	100	5	13.2 ± 12.4	64.1	100	5	9.2 ± 15.7 ^c^	66.1
	0	0	5.6 ± 6.3 ^a^	65.6	0	0	3.9 ± 5.0 ^a^	16.3
40 (V)	0	5	6.7 ± 8.5 ^a^	78.8	0	5	6.8 ± 13.5 ^a^	66.5
50 (R)	100	0	1.1 ± 1.4 ^b^	41.1	100	0	0.4 ± 0.9 ^b^	15.0
	100	5	5.6 ± 5.1 ^c^	71.7	100	5	7.2 ± 10.7 ^c^	54.0

V = vervet, R = rhesus, a, b, c, d = values in columns with different superscript letters were significantly different among the concentrations of CuSO_4_, CdCl_2_ and caffeine (*p* < 0.05).

## Data Availability

Not applicable.

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
