# Peer review of "Effect of Copper Sulphate and Cadmium Chloride on Non-Human Primate Sperm Function In Vitro"

_ijerph, 2021, doi:10.3390/ijerph18126200_

Round 1

Reviewer 1 Report

    As you said, you've reported on similar work before, "Andrologia, 2018, 50(10): e13141 ", and I've carefully compared this work to your previous work. I found that these two works basically described the similar conclusions, which studied the effects of CuSO4 and CdCl2 on the sperm motility and vitality, the only obvious difference was that the subjects were non-human primate. So, what's innovative about this article?

    I maintain that I have never seen an experimental error of this magnitude, which is incredible, such as “0.009 ± 0.033” and “-0.009 ± 0.025”. If you think it is feasible, please give examples of similar experimental results that have been published in reputable journals.

    In addition, how many experiments have been done to obtain the error range? Why not display the experimental data in the form of a line chart or bar chart with error bar.

Author Response

  1. “As you said, you've reported on similar work before, "Andrologia, 2018, 50(10): e13141 ", and I've carefully compared this work to your previous work. I found that these two works basically described the similar conclusions, which studied the effects of CuSO4 and CdCl2 on the sperm motility and vitality, the only obvious difference was that the subjects were non-human primate. So, what's innovative about this article?

Response:

Yes, as indicated before, the heavy metal concentrations used in this study were based on our published human study in which we found that these two metals had a detrimental effect on human sperm motility and vitality.

However, the purpose of the current study was to evaluate the suitability of the three non-human primate (NHP) species as research models for sperm functional testing. As mentioned in our Introduction, an animal model can only be “pharmacologically relevant for developmental and reproductive toxicity (DART) testing” if the test extract produces “a similar in vitro and in vivo pharmacokinetic effect as found in humans” (Subramanyam, M.; Rinaldi, N.; Mertsching, E.; Hutto, D. Selection of Relevant Species. In Preclinical Safety Evaluation of Biopharmaceuticals: A Science-Based Approach to Facilitating Clinical Trials; 2007); thus, the reason for using the same heavy metal concentrations as for human sperm to test the effect on our three NHP species’ sperm.

In addition, part of the innovation of the current study was to include the assessment of acrosome intactness and sperm hyperactivation as sperm functional parameters, which were not part of the human study.

Lastly, we made use of objective and standardized techniques, e.g. computer-aided sperm analysis, to assess sperm functional parameters. Not only do these techniques assist to accurately measure and recognize compromised sperm function, it is also vital in terms of validating the comparison of human and NHP results. It should also be highlighted that these techniques have not often been used in previous studies or publications on NHP sperm function, which further increases the novelty of our study.

  1. “I maintain that I have never seen an experimental error of this magnitude, which is incredible, such as “0.009 ± 0.033” and “-0.009 ± 0.025”. If you think it is feasible, please give examples of similar experimental results that have been published in reputable journals.”

Response:

We realize the results for the absorbance values at the 60 min reading are probably not feasible to report in the manuscript, since the cleavage reactions takes some time to be activated. Therefore, we removed the values for the 60 min reading in Table 2 and kept the 120 & 180 min readings. Significance was only found at the 180 min time point, as was indicated in the table and in the text.

  1. “In addition, how many experiments have been done to obtain the error range? Why not display the experimental data in the form of a line chart or bar chart with error bar.”

Response: For most of the results reported on in the manuscript between 3-7 repeats were done with semen samples from different males for each repeat. For the vitality assessment via absorbance readings discussed above, 4 repeats were included for the CuSO4 and 3 repeats for the CdCl2 (Table 2).

Reviewer 2 Report

The manuscript has improved and can be accepted for publication. Just small comments remain:

L244-248. We all know that PI means propidium iodide, but it would be good to add the abbreviation by maybe writing H&PI in the title of the chapter instead of just H&P. Likewise, in lines 343, 348 and 358, I would also write H&PI instead of H&P to keep consistency. Same comment for Table 1 legend should be considered and the discussion chapter.

L528. Remove the comma after “salt)”. Besides, in this line, it is written “formazan”, while in M&M it is written formazoan. Please, correct.

Author Response

  1. “L244-248. We all know that PI means propidium iodide, but it would be good to add the abbreviation by maybe writing H&PI in the title of the chapter instead of just H&P. Likewise, in lines 343, 348 and 358, I would also write H&PI instead of H&P to keep consistency. Same comment for Table 1 legend should be considered and the discussion chapter.”

Response:

Thank you for the suggestion to rather use “PI” as abbreviation for propidium iodide. We replaced “P” with “PI” in all sections of the manuscript where “P” indicated propidium iodide.

  1. “ Remove the comma after “salt)”. Besides, in this line, it is written “formazan”, while in M&M it is written formazoan. Please, correct.”

Response:

We corrected the text by deleting the comma in L528 and corrected the spelling of formazan in the M&M.

Reviewer 3 Report

The authors have made substantial improvement in the revised ms. It will be better just to organize in a better way the findings in order to be more understandable to readers

Author Response

  1. “The authors have made substantial improvement in the revised ms. It will be better just to organize in a better way the findings in order to be more understandable to readers.”

Response:

Since the Discussion section already has subheadings to indicate and organize the content, nothing was changed in this section (subheadings in Materials & Methods, Results and Discussion follow a similar pattern to make it easier to follow).

Part of the Conclusion was modified to clarify the purpose of the study as well as the main findings.

Round 2

Reviewer 1 Report

The authors did not fully answer the comments of reviewers, and the data with such large errors are not scientific enough to prove the conclusions obtained in the manuscript, so it is not recommended to publish.

Author Response

The authors agree with the reviewer that there is large variation in the absorbance values for both heavy metals investigated. We have explained in the discussion section that the large standard deviations reported in Table 2 are probably due to the heterogeneity of the samples used in this study. However, such large deviations cannot withstand scientific scrutiny.

In response to the reviewer’s concerns, Table 2 was removed from the text and added as a supplementary table.

The wording in section 3.2.2 was changed to indicate that the results obtained for the WST-1 cytotoxicity assay are only an estimate of the effect of the heavy metals on NHP sperm vitality and the resultant IC50 value for CuSO4.

This manuscript is a resubmission of an earlier submission. The following is a list of the peer review reports and author responses from that submission.

Round 1

Reviewer 1 Report

The authors evaluated the suitability and efficacy of sperm functional tests using NHP spermatozoa. In this regard, a concomitant aim was to evaluate the effect of CuSO4 and CdCl2 on selected sperm functions for three NHP species. Unfortunately, there are many serious problems that need to be addressed. I don’t think this work can be published:

  1. According to the concentration settings of the two heavy metals, the variation rules of the influence of these heavy metals on sperm activity could not be clearly found. Additionally, according to results of figures (such as Figure 1 (b,c,f), Figure 2 (a,b,d,e,f), Figure 3 (a-e)), in some cases it is even possible to suggest that low concentrations of heavy metal ions can promote some indicators of sperm activity.
  2. In the tables of the article, "±" should be expressed as the error range of this value, which is too bad, the error range will become so large that the data results have no credibility at all.
  3. The authors should study or speculate on how the two heavy metals affect sperm motility, rather than simply changing the heavy metal dose to get some intuitive experimental results. Make this article an academic research paper, not an experimental report.

Reviewer 2 Report

The present manuscript aims to evaluate the effect of copper sulphate and cadmium chloride in non-human primate sperm function in vitro. The manuscript is interesting, but some comments/suggestions follow:

The authors state that VCL was used to separate the ejaculates into three different subpopulations. That arises a couple of questions from this reviewer:

  1. Why these specific values (5<80>120)? Can the authors provide some literature about it or it’s just their own experience
  2. Why did the author use only VCL to establish sperm motile subpopulations? There are hundred of manuscripts which explain that motile subpopulations are statistically calculated from all motile parameters. For example, by applying the VARCLUS procedure if you work with SAS package.

L278. Why did the authors set P≤0.05? It is always set at P<0.05. In addition, this P value changes along the text. Sometimes, the authors state it was ≤0.05, but P<0.05 has been stated in some sentences. Please, correct.

In the chapter of motility results, the authors state that the presence of heavy metals induces a significant decrease in motility parameters. While this totally true, a significant decrease is also observed in control groups, which I assume had no heavy metal in the dilution. This has to be clarified in the text. In fact, the important result would be to determine if there are differences between control-samples and metal-samples.

According to Table 1, semen from vervet monkeys was stained with H&P and semen from baboon monkeys was stained with E-N. Was not semen from vervet monkeys evaluated also with H&P?

L423-424. This sentence is incomprehensible. Please, re-write.

L429-431. The authors state that H&P test needs a long incubation period to detect a change in vervet sperm viability. According to this sentence, one can understand that the activity of the test depends on the time of incubation, when the most probable scenario is that a long period incubation is needed to induce alterations in sperm cells viability. Please, re-write the sentence.

SMALL COMMENTS

L31. It looks like there is an extra space between the words “as” and “plastics”

L101. I guess the “S” at the end of the sentence is a mistake.

L164. Remove the extra space between CuSO4 and the bracket.

Extra spaces have been detected along the text. Please, correct.

Finally, some language/gramma mistakes have been detected in the manuscript. Please, check.

Reviewer 3 Report

The authors tried to find any correlation between metals and sperm function hoping these results to extrapolate to humans.

The authors should use updated references in the Introduction section, Int J Environ Res Public Health. 2018 May 30;15(6):1117 and Toxics. 2017 Dec 21;6(1):2 elaborating more in the impact of enviromental factors in male fertility.

Since all the incubation time and the concentrastions used had detrimental effects on sperm parameters which concentration is affordable for sperm motility. This is very crucial since the use of metals in every day may be harmful or not.

Reviewer 4 Report

The authors investigate the effect of copper sulfate and cadmium chloride on sperm quality of vervet monkeys and chacma baboons. In addition, one rhesus monkey was used for sperm motility and hyperactivation tests. The objective of this study was to investigate whether non-human primates can be used as a research model for toxicological studies of cadmium and copper effects on men reproduction, although the authors previously found  that both heavy metals show negative effects on human sperm (PMID: 30225848, DOI: 10.1111/and.13141). Using an experimental design similar to the one employed in their previously published article, the authors confirm that both metals have a negative effect on sperm parameters in non-human primates. Therefore, the study provides limited aspect of originality and novelty. There are also flaws in the materials and methods: for instance, 100 spermatozoa were evaluated per acrosome and membrane integrities instead of 200 sperm recommended by the WHO. This, together with the low sample size, may have contributed to the large standard deviation of results. Also, sperm samples from three males (one for each species) were analyzed by the Computer Assisted Sperm Analyzer, which is not sufficient to draw conclusions. There are missing results (e.g. slow, medium, and rapid populations or WST-1 at 4 hours of incubations despite their description in the material and methods section). Viability and vitality tests are redundant since both tests assess the same sperm parameter (i.e. plasma membrane integrity) in contrast to what stated in lines 413-414.

The authors should also have cited the following publication:

Farren Chelsea Prag. Evaluation of standard and development of new sperm functional tests in selected primate species. Master thesis. University of Cape Town. 2017. Available online at http://etd.uwc.ac.za/handle/11394/5673